# Effects of Betahistine on the Development of Vestibular Compensation after Unilateral Labyrinthectomy in Rats

**DOI:** 10.3390/brainsci11030360

**Published:** 2021-03-11

**Authors:** Junya Fukuda, Kazunori Matsuda, Go Sato, Tadashi Kitahara, Momoyo Matsuoka, Takahiro Azuma, Yoshiaki Kitamura, Koichi Tomita, Noriaki Takeda

**Affiliations:** 1Department of Otolaryngology, Graduate School of Biomedical Sciences, University of Tokushima, Tokushima 770-8503, Japan; go-sato@tokushima-u.ac.jp (G.S.); momomonga_buchi@yahoo.co.jp (M.M.); azuma.takahiro@tokushima-u.ac.jp (T.A.); ykitamura@tokushima-u.ac.jp (Y.K.); takeda@tokushima-u.ac.jp (N.T.); 2Department of Otolaryngology, Tokushima Prefectural Central Hospital, Tokushima 770-8503, Japan; matsudakazunori1024@gmail.com; 3Department of Otolaryngology Head and Neck Surgery, Nara Medical University, Kashihara 634-8522, Japan; tkitahara@naramed-u.ac.jp; 4Department of Anatomy and Developmental Neurobiology, Institute of Biomedical Science, Graduate School, Tokushima University, Tokushima 770-8503, Japan; ktomita@tokushima-u.ac.jp

**Keywords:** vestibular compensation, betahistine dihydrochloride, medial vestibular nucleus, Fos, histamine H3 receptor

## Abstract

Background: Vestibular compensation (VC) after unilateral labyrinthectomy (UL) consists of the initial and late processes. These processes can be evaluated based on the decline in the frequency of spontaneous nystagmus (SN) and the number of MK801-induced Fos-positive neurons in the contralateral medial vestibular nucleus (contra-MVe) in rats. Histamine H3 receptors (H3R) are reported to be involved in the development of VC. Objective: We examined the effects of betahistine, an H3R antagonist, on the initial and late processes of VC in UL rats. Methods: Betahistine dihydrochloride was continuously administered to the UL rats at doses of 100 and 200 mg/kg/day using an osmotic minipump. MK801 (1.0 mg/kg) was intraperitoneally administered on days 7, 10, 12, and 14 after UL, while Fos-positive neurons were immunohistochemically stained in the contra-MVe. Results: The SN disappeared after 42 h, and continuous infusion of betahistine did not change the decline in the frequency of SN. The number of MK801-induced Fos-positive neurons in contra-MVe significantly decreased on days 7, 10, and 12 after UL in a dose-dependent manner in the betahistine-treated rats, more so than in the saline-treated rats. Conclusion: These findings suggest that betahistine facilitated the late, but not the initial, process of VC in UL rats.

## 1. Introduction

Unilateral vestibular dysfunction causes spontaneous nystagmus (SN), changes in postural control, and locomotor deviation. SN is a high-frequency, mainly horizontal, spontaneous ocular nystagmus, with its quick phase directed to the unaffected side. However, the nystagmus and balance disorders recover gradually after the occurrence of the lesion. This functional restoration—based on the plasticity of the central vestibular system—is called vestibular compensation (VC) [1]. The VC is divided into two phases: static and dynamic. Static VC is further divided into the initial and late processes [2]. Unilateral labyrinthectomy (UL) causes a loss of resting activity in many neurons in the ipsilateral medial vestibular nucleus (ipsi-MVe), resulting in an imbalance in neural activity between the vestibular nuclei on each side of the brainstem. This imbalance in vestibular nuclear activities after UL induces SN. During the initial process of VC after UL, the suppression of the contralateral medial vestibular nucleus (contra-MVe) by the vestibular cerebellum–vestibular nucleus inhibitory system equalizes the imbalance in vestibular nuclear activities. Consequently, the SN gradually decreases [2,3]. During the late process of VC after UL, the spontaneous firing of ipsi-MVe neurons is restored by changes in cell membrane properties, reinforcing the balance in neural activity between the vestibular nuclei on each side. Consequently, the neural activities between the vestibular nuclei on each side are balanced without the vestibular-cerebellum-induced suppression of the contra-MVe during the initial VC process [4,5].

Betahistine is a histamine H3 receptor (H3R) antagonist and a partial histamine H1 receptor agonist. Betahistine is clinically used for the treatment of vertigo and Meniere’s disease-like symptoms, and its anti-vertiginous action has been considered to be in part due to its increase of inner ear microcirculation via H1R. A previous study reported that betahistine accelerated the disappearance of SN after UL in cats, suggesting that betahistine facilitates VC after UL via H3R [6]. The aim of the present study was to investigate the effects of betahistine on the initial and late processes of VC, using unilaterally labyrinthectomized rats (UL rats) as model animals. To evaluate the effects of betahistine on the initial process of VC in UL rats, we measured the decline in the frequency of SN after UL [7]. The late process of VC in UL rats can be evaluated by measuring the decline in the number of Fos-positive neurons being expressed in the contra-MVe after induction by MK801, an *N*-methyl-D-aspartate (NMDA) receptor antagonist [8]. Therefore, we investigated the effects of betahistine on the number of MK801-induced Fos-positive neurons in the contra-MVe of UL rats.

## 2. Materials and Methods

### 2.1. Experimental Animals

Experiments were performed using adult male Wistar rats (Japan SLC, Inc., Hamamatsu City, Shizouka, Japan) weighing approximately 150–200 g. The rats were individually housed in polycarbonate cages with wood chip bedding at an environmental temperature of 20–22 °C under a 12 h light/dark cycle (lights on at 08:00, lights off at 20:00), and were provided unlimited access to water and food. This study was conducted in accordance with the Fundamental Guidelines for Proper Conduct of Animal Experiments and Related Activities in Academic Research Institutions under the jurisdiction of the Ministry of Education, Culture, Sports, Science, and Technology, Japan. All procedures were approved by the Division for Animal Research Resources and Genetic Engineering Support Center for Advanced Medical Sciences, Institute of Biomedical Sciences, Tokushima University Graduate School (Animal Experiment Plan Approval No. T2019-9).

### 2.2. Unilateral Labyrinthectomy

The animals were anesthetized with isoflurane (Wako Pure Chemical Industries, Ltd., Osaka City, Osaka, Japan). The UL of the right ear was performed as previously described [9]. In brief, the tympanic bulla was opened under an operating microscope. The tympanic membrane, incus, and malleus were subsequently removed using a postauricular approach. The foot plate of the stapes was removed to open the oval window. The membranous labyrinth was destroyed by injecting 100% ethanol. An antibiotic (ofloxacin) cream was applied to the opened tympanic cavity to prevent infection at the end of surgery. Finally, the incision was closed, after which the animals were allowed to recover in light. UL was confirmed by the appearance of SN and postural deviation after recovery from anesthesia. The sham-operated rats were subjected to only a right postauricular skin incision under anesthesia.

### 2.3. Drug Administration

MK801 (Santa Cruz Biotechnology, Inc., Santa Cruz, CA, USA) was dissolved in 0.9% saline and injected intraperitoneally at a dose of 1.0 mg/kg in rats 7, 10, 12, and 14 days after UL. The MK801 dose of 1.0 mg/kg was used as it has been reported to induce complete decompensation in UL rats [2].

Betahistine dihydrochloride (Sigma-Aldrich, St. Louis, MO, USA) was dissolved in 0.9% saline and infused intraperitoneally (0.5 µL/h) at a dose of 100 mg/kg/day or 200 mg/kg/day using an osmotic minipump (Alzet, Palo Alto, CA, USA) immediately after UL. The osmotic minipump was filled with either saline or betahistine dihydrochloride and implanted intraperitoneally until day 14 after UL. These doses were chosen because betahistine dihydrochloride administered at a dose of 100 mg/kg/day has been reported to induce significant improvements in posture recovery in cats after UL [10].

### 2.4. Behavioral Examination

The eye movements were recorded using a video camera (Sony HDR-CX500V, Sony Corporation, Minato City, Tokyo, Japan) with a zoom lens. The frequency of SN in UL rats was counted as the number of quick phase beats in 15 s. The video images were recorded on a memory card and replayed on a liquid crystal display screen. The frequency of SN was counted three times for each rat, and the average value was calculated as previously described [8]. The values were measured at 0.5, 1, 2, 3, 6, 12, 18, 24, 30, 36, and 42 h after UL.

### 2.5. Tissue Preparation and Immunohistochemical Staining

Tissue preparation and immunohistochemical staining were performed as described previously [8]. The rats were deeply anesthetized with isoflurane 2 h after intraperitoneal administration of MK801. Following this, they were perfused transcardially with 100 mL of 4 °C saline and then with 250 mL of 4% paraformaldehyde in 0.1 mol/L phosphate buffer (PB). The rat brain was promptly extracted after perfusion—fixation and post-fixed in the same fixative solution at 4 °C for 1 day. The fixed brains were submerged in 30% sucrose PB at 4 °C for 2 days. The tissues were frozen and sectioned to a thickness of 30 µm using a cryostat (Leica CM1850, Leica Biosystems, Wetzlar, Germany), then the immunohistochemical reaction was visualized using the peroxidase—antiperoxidase method. The sections were pretreated with 0.1% H_2_O_2_ in 0.3% Triton X-100 in phosphate-buffered saline (PBS) for 30 min and then incubated in 5% normal goat serum (NGS) in PBS containing 0.3% Triton-X (PBST). The sections were then incubated with rabbit polyclonal anti-Fos antibody (Santa Cruz Biotechnology) diluted to 1:5000 in PBST containing 1% NGS for 2 days at 4 °C. Following incubation with the primary antibody and a brief wash, the sections were incubated with anti-rabbit IgG (Medical and Biological Laboratories Co., Ltd., Nagoya, Japan) diluted to 1:1000 in PBST for 1 h at room temperature. Fos expression in neurons was induced by incubating sections with a 3,3’-diaminobenzidine (DAB) Substrate Kit (Vector Laboratories, Burlingame, CA, USA).

### 2.6. Cell Counting

Transverse 30-µm-thick sections of the brainstem were observed under bright-field microscopy at 40× and 100× magnification to detect Fos-positive neurons in the contra-MVe. Only cells that had significant levels of DAB reaction product in their nuclei (i.e., above tissue background levels) were counted using a digital image analysis system (ImageJ ver. 1.52a software).

### 2.7. Experimental Protocol

Depending on the operative and postoperative treatments, the experimental rats were divided into four groups: (1) rats that received saline after UL, (2) rats that received betahistine dihydrochloride (100 mg/kg/day) after UL, (3) rats that received betahistine dihydrochloride (200 mg/kg/day) after UL, and (4) sham-operated rats. Rats that received saline were subjected to UL of the right ear and received intraperitoneal injection of saline with an osmotic minipump (n = 19). Then, they were intraperitoneally administrated a single dose of MK801 (1.0 mg/kg) on days 7 (n = 4), 10 (n = 5), 12 (n = 5), and 14 (n = 5) after UL.

Rats that received betahistine dihydrochloride were subjected to UL of the right ear and received intraperitoneal infusion of betahistine dihydrochloride at a dose of 100 mg/kg/day (n = 21) or 200 mg/kg/day (n = 20) with an osmotic minipump. Rats that received 100 mg/kg/day betahistine dihydrochloride were then intraperitoneally administered a single dose of MK801 (1.0 mg/kg) on days 7 (n = 6), 10 (n = 7), 12 (n = 4), and 14 (n = 4) after UL. Rats that received 100 mg/kg/day betahistine dihydrochloride were also intraperitoneally administered a single dose of MK801 (1.0 mg/kg) on days 7 (n = 4), 10 (n = 7), 12 (n = 4), and 14 (n = 5) after UL. The sham-operated rats were intraperitoneally administered a single dose of MK801 (1.0 mg/kg) on day 14 after the sham operation (n = 4).

### 2.8. Statistical Analysis

The frequency of SN and the number of Fos-positive neurons were analyzed using a two-factorial analysis of variance followed by the Shceffé’s test. Statistical significance was set at *p* < 0.05.

## 3. Results

The frequency of SN was measured 0.5, 1, 2, 3, 6, 12, 18, 24, 30, 36, and 42 h after UL (Figure 1). At each time point, the SN frequency values were not significantly different between the UL rats that received betahistine (100 or 200 mg/kg/day) and those that received saline. The SN disappeared at 42 h in all UL rats that received either betahistine or saline.

MK801 induced the appearance of Fos-positive neurons in the contra-MVe in UL rats. Figure 2 shows the substantial number of MK801-induced Fos-positive neurons in the contra-MVe on day 10 after UL in saline-treated rats. In rats that received saline, the number of MK801-induced Fos-positive neurons in the contra-MVe gradually decreased and equaled the number in the sham-operated rats on day 14 after UL (Figure 3). The number of MK801-induced Fos-positive neurons in the 100 mg/kg/day betahistine-treated rats was significantly lower than that in the saline-treated UL rats on days 7, 10, and 12 (Figure 3), and equaled the number in the sham-operated rats on day 12 after UL. The number of MK801-induced Fos-positive neurons in the 200 mg/kg/day betahistine-treated rats was also significantly lower than that in the saline-treated UL rats on days 7, 10, and 12, but equaled the number in the sham-operated rats on day 10 after UL. Moreover, the number of MK801-induced Fos-positive neurons on day 10 after UL was significantly lower in the rats that received 200 mg/kg/day betahistine than in those that received a 100 mg/kg/day dose (Figure 3).

## 4. Discussion

In the present study, the decline in the frequency of SN after UL was not different between the betahistine- and saline-treated rats. In both groups, the SN disappeared 42 h after UL. Because the decline in the frequency of SN after UL indicates the development of the initial process of VC in rats [7], our findings suggest that betahistine did not affect this process of VC after UL in these rats.

A previous study reported that the disappearance of SN after UL was accelerated in cats that received betahistine transorally [6]. A recent study reported that betahistine microinjected into the bilateral medial vestibular nuclei also decreased the frequency of SN in the UL rats more so than in the control rats [11]. However, our previous study showed that intraperitoneal infusion of thioperamide (another H3R antagonist) did not change the frequency of SN after UL in rats [8]. Moreover, intraperitoneal injection of thioperamide has been reported to delay the recovery of body tilt after UL in goldfish [12]. Thus, the effects of H3R antagonists—including betahistine—on the initial process of VC varies with species and administration methods.

In the present study, the number of MK801-induced Fos-positive neurons in the contra-MVe of betahistine-treated UL rats decreased in a dose-dependent manner and was lower than that in the saline-treated UL rats. The numbers of such neurons in the saline-treated, 100 mg/kg/day betahistine-treated, and 200 mg/kg/day betahistine-treated UL rats equaled those in the sham-operated rats on days 14, 12, and 10 after UL, respectively. As the decline in the number of MK801-induced Fos-positive neurons in the contra-MVe after UL indicates the development of the late process of VC in rats [8], these findings suggest that betahistine facilitates the late process of VC after UL in rats in a dose-dependent manner.

Because Fos is the protein product of an immediate early gene, c-Fos is a marker of neural activity [13]. It is assumed that the MK801-induced Fos-positive neurons in the contra-MVe are suppressed by the NMDA receptor-mediated vestibular cerebellum-vestibular nucleus inhibitory system in the initial process of VC after UL [14]. Our previous study showed that the MK801-induced Fos-positive neurons in the contra-MVe disappeared in accordance with the recovery of the spontaneous firing of the ipsi-MVe neurons in the late process of VC after UL [8]. Therefore, the present findings further suggest that betahistine facilitates recovery of the spontaneous firing of the ipsi-MVe neurons in UL rats.

H3Rs are located in the presynaptic histaminergic fibers and act as inhibitory autoreceptors [15]. The histaminergic neurons in the tuberomammillary nuclei of the posterior hypothalamus give rise to the axons to the MVe [16]. Therefore, it is assumed that H3R antagonists increase the release of histamine in the MVe. Indeed, betahistine, an H3R antagonist, has been reported to upregulate the gene expression of histidine decarboxylase, a histamine-synthesizing enzyme in the tuberomammillary nuclei of cats, suggesting that betahistine upregulates histamine turnover and release [17]. Moreover, histamine H1, H2, and H3 receptors (H1R, H2R, and H3R) are also expressed in the MVe of rats [3,18], while H1R- and H2R-mediated histamine cause excitation in the MVe neurons in vivo and in vitro [19,20,21]. Therefore, it is suggested that the H3R antagonist acts by increasing histamine release and causing excitability in the MVe neurons. Betahistine works as both a partial H1R agonist and a more potent H3R antagonist [22,23], while H1R expression increases in a restricted manner in the ipsi-MVe neurons of UL rats [11]. Therefore, it is suggested that the betahistine-induced release of histamine causes the activation of the MVe neurons and facilitates the recovery of the spontaneous firing of the ipsi-MVe neurons. It is further suggested that the neural activities between the vestibular nuclei on each side are finally balanced in the late VC process, without the vestibular cerebellum-induced suppression of the contra-MVe that occurs during the initial VC process. Consequently, the late VC process was facilitated in the UL rats that received betahistine.

Clinically, betahistine is usually used as an anti-vertiginous drug for the treatment of acute and subacute stages of vertigo. Betahistine has been reported to improve the recovery of static vestibular symptoms in patients with Meniere’s disease after unilateral vestibular nerve transection [24]. This clinical finding suggests that betahistine may also facilitate the development of VC after unilateral vestibular deficits in humans. Based on the present finding using betahistine at doses of 100–200 mg/kg/day in rats and previous reports using betahistine at doses of 50–100 mg/kg/day in cats [10], our clinical goal is to prove the hypothesis that a high dose of betahistine facilitates the process of VC in patients with unilateral vestibular deficits, who complain of refractory dizziness due to delays in development of VC. In fact, our preliminary study showed long-term administration of betahistine at a high dose of 72 mg/day for 16 weeks improved Dizziness Handicap Inventory scores in refractory dizzy patients with vestibular neuritis. Although the original daily maximum dose of betahistine was 48 mg, it was recently reported that long-term administration for 12 months with betahistine at higher dose, such as 144 mg/day and more recently 480 mg/day, was effective for the treatment of Meniere’s disease [25,26]. Further studies are needed on whether log-term treatment with a high dose of betahistine facilitates the process of VC in refractory dizzy patients with unilateral vestibular deficits.

## 5. Conclusions

In the present study, we evaluated the effects of betahistine on the initial and late processes of VC in UL rats by measuring the decrease in the frequency of SN and the decline in the number of MK801-induced Fos-positive neurons in the contra-MV, respectively. Betahistine did not affect the initial VC process, but facilitated the development of the late VC process in UL rats. Our results suggest that betahistine blocks presynaptic H3R and promotes histamine release. This, in turn, activates the MVe neurons via H1Rs and H2Rs to facilitate the recovery of the spontaneous firing of the ipsi-MVe neurons, accelerating the late process of VC after UL in rats. The facilitatory effect of betahistine on VC is an underlying mechanism in the management of acute and subacute vertigo in patients with unilateral vestibular disorders.

## Figures and Tables

**Figure 1 brainsci-11-00360-f001:**
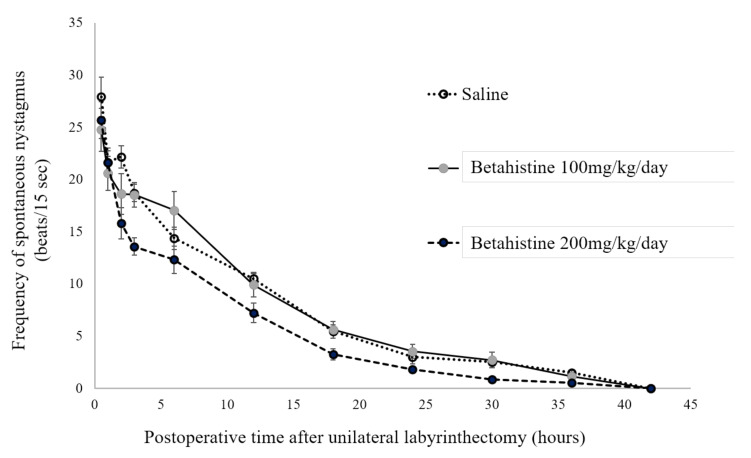
Effects of betahistine on changes with time in the frequency of spontaneous nystagmus after unilateral labyrinthectomy in rats. Each point represents the mean number of quick-phase eye movements in 15 s. Saline: rats received intraperitoneal infusion of saline, shown as open circles with a dotted line. Betahistine 100 mg/kg/day: rats received intraperitoneal infusion of betahistine at a dose of 100 mg/kg/day, shown as gray circles with a solid line. Betahistine 200 mg/kg/day: rats received intraperitoneal infusion of betahistine at a dose of 200 mg/kg/day, shown as filled circles with a broken line.

**Figure 2 brainsci-11-00360-f002:**
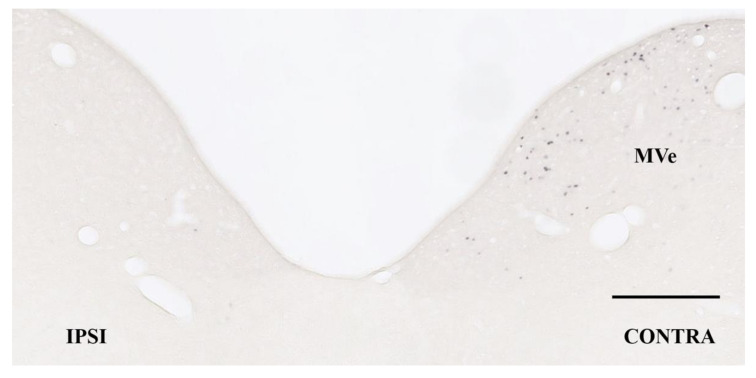
Bright-field micrograph of Fos-positive neurons in the contralateral medial vestibular nucleus 2 h after administration of MK801 in unilateral labyrinthectomized rats who received continuous intraperitoneal administration of saline on post-operative day 10 after unilateral labyrinthectomy (UL). Bar: 200 µm. MVe: medial vestibular nucleus; IPSI: ipsilateral to UL; CONTRA: contralateral to UL.

**Figure 3 brainsci-11-00360-f003:**
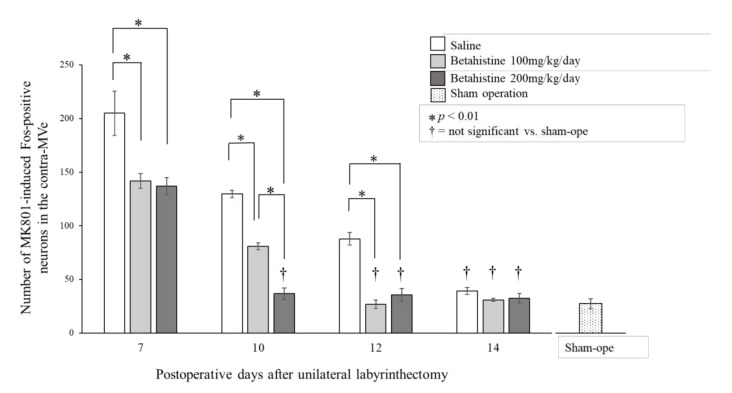
Effects of betahistine on changes with time in the number of Fos-positive neurons in the contralateral medial vestibular nucleus 2 h after administration of MK801 in unilateral labyrinthectomized rats. Saline: rats received intraperitoneal infusion of saline, shown as open columns. Betahistine 100 mg/kg/day: rats received intraperitoneal infusion of betahistine at a dose of 100 mg/kg/day, shown as gray columns. Betahistine 200 mg/kg/day: rats received intraperitoneal infusion of betahistine at a dose of 200 mg/kg/day, shown as filled columns. Sham operation (sham-ope): rats received sham operation only.

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
