# Peer review of "Effects of Betahistine on the Development of Vestibular Compensation after Unilateral Labyrinthectomy in Rats"

_brainsci, 2021, doi:10.3390/brainsci11030360_

Round 1

Reviewer 1 Report

The manuscript appears very interesting presenting an interesting study on rats. I recommend some changes to make the manuscript more publishable. First of all expand the introductory chapter, explaining in more detail about the spontaneous nystagmus and the effect of betaistin The really problem of the manuscript is that you do not add nothing of original to literature In the final chapter it would be better to make the goal of the study more compelling. the reader must understand why betaistin in important during labirintectomy, and the right posology. finally, a revision of the English language by a native speaker technician.  

Reviewer 2 Report

Clinically, this drug is in use worldwide both for vestibulopathy and Meniere's disease. But there is still issue regarding the effective concentration clinically. Therefore, providing the comparison of clinical dosage and the concentration used in the present study would help the readers to apply this work in clinics. There are incomplete sentences at the end of figure legends 'is the figure'. Please double check.
